# Adjuvant Radiation in Resectable Node-Positive Merkel Cell Carcinoma in the Immunotherapy Era: Implications for Future and Ongoing Trials

**DOI:** 10.3390/cancers15235550

**Published:** 2023-11-23

**Authors:** Paul Riviere, Anna M. Dornisch, Parag Sanghvi, Loren K. Mell

**Affiliations:** 1Department of Radiation Medicine and Applied Sciences, University of California San Diego, La Jolla, CA 92093, USA; 2Center for Health Equity and Education Research (CHEER), University of California San Diego, La Jolla, CA 92093, USA

**Keywords:** radiation therapy, immunotherapy, Merkel cell carcinoma, stage migration

## Abstract

**Simple Summary:**

Merkel cell carcinoma (MCC) is a rare, aggressive type of skin cancer that has been found to respond well to immunotherapy in the metastatic setting; ongoing large trials are exploring the addition of this therapy to earlier stage disease to prevent recurrence after surgery, particularly distant metastatic recurrence. However, newer clinical data suggest that non-metastatic MCC may have a better prognosis than what was previously observed, in part because of new technologies that have improved the detection of MCC spread to regional lymph nodes or other organs (i.e., distant metastases). Thus, preventing MCC from recurring in the area of the original surgery may be more important in achieving a cure for these patients without distant metastatic disease. Radiation therapy is an effective treatment for preventing these local recurrences, but there is significant variability in how ongoing immunotherapy trials allow for its use. This review describes the evidence supporting the ongoing use of adjuvant radiation therapy, how variability in radiation therapy use may have led to negative results in a large trial of immunotherapy in non-metastatic MCC, and the risk this poses for two large ongoing studies in this domain.

**Abstract:**

Merkel cell carcinoma (MCC) is a cutaneous malignancy often treated with surgical resection followed by adjuvant radiation therapy (RT). In the node-positive setting, adjuvant RT reduces the risk of locoregional recurrence, but historical data suggest that distant failure is a persistent issue and often fatal. This has prompted new efforts to intensify treatment in these patients with the addition of neoadjuvant or adjuvant immune checkpoint inhibitor therapy. However, newer diagnostic techniques have led to stage migration in patients with previously subclinical metastatic disease; consequently, preventing locoregional recurrence may be a higher priority in node-positive MCC patients than was previously believed. Recent trials in node-positive MCC, such as ADMEC-O, have had lower rates of adjuvant RT utilization in treatment versus control arms, which may have attenuated the observed effect of adjuvant immunotherapy. The low utilization of adjuvant RT may have also resulted in a higher recurrence rate in patients who did not have a complete response to neoadjuvant immunotherapy in the CHECKMATE 358 trial. Altogether, these are important considerations for ongoing and future immunotherapy trials in MCC and may affect the interpretation of their results. Ongoing clinical trials may determine which patients are at low risk of recurrence when treated with immunotherapy and whether adjuvant RT could be omitted in select patients.

## 1. Introduction

Node-positive Merkel cell carcinoma (MCC) has a high rate of locoregional recurrence when not treated with adjuvant radiation therapy (RT) (Table 1) [1,2,3,4,5,6,7,8]. Importantly, historical case series have also found that these patients may be at a high risk of distant metastatic recurrence [8], which is the principal driver of cancer-related mortality [9]. Given that, in the metastatic setting, MCC has an excellent response rate to checkpoint inhibitor immunotherapy [10], there is increasing interest in the addition of systemic immunotherapy in patients with node-positive or high-risk early stage MCC.

There are two general immunotherapy approaches under investigation: adjuvant and neoadjuvant; each theoretically has specific advantages and disadvantages. Trialists have hypothesized that, given the excellent initial response rate to MCC, a brief course of neoadjuvant immunotherapy could downsize tumors and improve the control of distant disease. However, this approach also increases the risk of tumor progression in immunologically cold tumors, and could introduce delays that render a patient inoperable or otherwise unable to undergo curative intent therapy [20]. In other high-risk cutaneous malignancies, studies of neoadjuvant immunotherapy have begun to explore whether complete responses to checkpoint immunotherapy could be used to omit adjuvant RT in select patients [21]. In contrast, adjuvant immunotherapy would allow operable patients to proceed immediately to curative-intent local therapy, but may require a longer course of treatment, with a higher likelihood of toxicity, greater treatment cost, and a risk of overtreatment [22].

As these paradigms evolve, there remains compelling evidence to support the use of adjuvant RT for local consolidation. First, data suggest that there has been significant stage migration in node-positive MCC and that distant metastatic disease may not be as frequent as previously thought in the era of PET/CT and sentinel lymph node biopsy (SLNB). This may result in immunotherapy having a smaller than expected absolute reduction in distant failure; whether immunotherapy has a significant additive effect or is able to replace radiotherapy for local consolidation is uncertain. Not accounting for these effects poses risks for ongoing trials; recently, a large phase II study failed its primary endpoint, possibly in part due to the decreased utilization of adjuvant radiation in the immunotherapy arm compared with the observation arm [22]. Second, in the adjuvant setting, protracted courses of immunotherapy are associated with significant toxicity, most frequently cutaneous/fatigue, but also including more severe gastrointestinal, pulmonary, or endocrinological effects [22]. Third, patients receiving neoadjuvant immunotherapy who do not have a pathologic complete response appear to have a high rate of recurrence, and the de-escalation of adjuvant RT in patients who have had a pathological complete response has not yet been formally evaluated.

## 2. Radiation and Local Consolidation

To date, there have been no completed randomized trials of adjuvant RT in the localized or node-positive MCC setting. Much of the data guiding recommendations are derived from retrospective data, often from single institutions (Table 1) [23,24]. Guidelines generally recommend doses of 50–60 Gray for adjuvant radiation (or higher with grossly positive margins) [24], which is consistent with most reported case series [5,7,8,11,19], though details on doses and volumes are inconsistently reported and are often not reported at all [25]. While there are studies examining hypofractionated [26] or dose-de-escalated [27] regimens for patients with a high burden of competing comorbid conditions, the focus of this study is on conventionally fractionated regimens, which remain the standard practice in radiation oncology. Even recently published series often include patients diagnosed over several decades, prior to the standard utilization of SLNB or PET/CT for staging. In this setting, node-positive patients treated with adjuvant RT generally had a low rate of local recurrence (as low as 12% in one series; Table 1) [12], but relatively high rates of recurrence overall: one study (with variable use of adjuvant RT) found almost a 70% rate of overall recurrence in patients with clinically node-positive disease [6]. Indeed, an NCDB analysis found that, in the node-positive setting, the use of adjuvant RT was not associated with an effect on overall survival [1] (though these results are in contrast to a smaller SEER analysis which did find a benefit in overall survival) [28]. While comparative effectiveness research in such retrospective databases is generally challenging to interpret [29], the difficulty in proving an overall survival benefit in older studies is consistent with the observation that rapid progression with distant metastases is the principal cause of cancer-related mortality in the node-positive setting [9]. However, as will be discussed below, stage migration has significantly affected patterns of failure with node-positive, non-metastatic MCC. The most contemporary randomized trial reporting patterns of failure in this population identified locoregional failure as the predominant site of recurrence (approximately two-thirds of recurrences) [22]. Modern studies will be necessary to determine whether adjuvant RT confers a survival advantage, now that previously subclinical nodal or metastatic disease can be identified at the time of staging.

Independent of the question of overall survival, RT is associated with several important benefits beyond reducing locoregional recurrence. In the node-positive setting, nodal irradiation may reduce the need for node dissection, particularly in patients with microscopic nodal disease identified on SLNB [2,11,12]. Additionally, adjuvant RT appears to obviate the need for wide (or even microscopically negative) surgical margins [7], which is particularly relevant for primary tumors of the head and neck, where a larger excision could result in a functional or cosmetic defect. In the head and neck region specifically, SLNB may be less reliable [23,24], and patients have a high rate of locoregional recurrence. One study found that, even in ostensibly low-risk disease of the head and neck, adjuvant RT was associated with a reduced risk of locoregional recurrence from 26% to 0% (*p* = 0.02) [30].

Few modern studies have formally reported toxicity specifically in MCC patients, but acute grade ≥ 3 toxicity generally occurs in 5–10% of patients and is predominantly cutaneous (or mucositis with head and neck primary) [31,32]; chronic grade ≥ 3 toxicity is rare. It is likely that there is regional anatomical variation in the risk of lymphedema with nodal radiation: in the axilla, RT likely adds less than 5% absolute risk of long-term lymphedema to a SLNB [33], and in the groin region, it increases this risk at 6 months by 11% [34]. Generally, adjuvant RT is associated with a lower rate of lymphedema than nodal dissection [34,35].

## 3. Stage Migration

Critical to the understanding of contemporary immunotherapy trials in MCC is the issue of stage migration. Many trials and case series predate the systematic use of SLNB and PET/CT for staging, and even more recently published series include patients from the pre-SLNB and PET/CT era. Of the trials in Table 1, Hui et al. [8] only used PET/CT to stage 24% of patients, Broida et al. reported 35% [17], and Deneve et al. [14] reported that PET/CT was used, but did not specify the frequency; the other trials did not include PET/CT staging or did not report on its use. These technologies have upstaged many stage I–II patients to stage III through the radiographic and/or pathologic detection of subclinical nodal disease, while also upstaging stage III patients to stage IV with the detection of subclinical distant metastatic disease. Thus, relative to prior eras, the contemporary pool of stage III patients includes patients with a more favorable prognosis [36], which partially explains the excellent outcomes in the observation arm of the more modern ADMEC-O trial (discussed below) [22] compared with older series.

There are several studies that demonstrate stage migration in MCC. One study reported that, among 194 MCC patients without initial distant metastatic disease (AJCC stage I–III) [37] who subsequently developed metastases, 51% of patients did so within a year of definitive local therapy [9]. Though this study did not specify the use of staging imaging at diagnosis, a separate publication from the same registry with patients diagnosed in the same time range reported that 14.6% of patients had no staging imaging and 40% of patients with imaging did not have a staging PET [38]. While it is possible that the observed short time to distant metastatic failure in patients with ostensibly localized disease is the result of aggressive recurrent disease, it is likely that many of these patients with rapid progression had undetected prior metastatic disease.

PET/CT performs well in MCC staging, with an estimated sensitivity of 90% and specificity of 98% in meta-analyses [37]. Studies have reported that PET changes the management of over one-third of patients with a new diagnosis of MCC [39] and upstages about 17% of patients [38,39]. In the clinically node-positive setting, PET may detect previously occult metastatic disease in up to 50% of patients [40]. Importantly, PET/CT appears to upstage patients at a higher rate than conventional CT alone (18.8% versus 6.9%, *p* = 0.006 in one study) [38].

In addition to PET/CT staging, SLNB has been a critical addition to guideline-directed MCC staging in clinically node-negative patients. SLNB has been reported to detect occult nodal disease in 26% [41] to 42% [42,43] of patients. One study reported SLNB detection of nodal disease in as many as 31% of patients with radiographically and clinically node-negative disease [38].

Taken together, PET/CT staging (not reported but likely performed systematically) [23] and SLNB or lymph node dissection (performed in 66% and 44% of patients, respectively) could, in part, explain how the node-positive patients in the observation arm of the ADMEC-O study [22] achieved a 2-year disease-free survival rate of 67.3%. It should be noted that adjuvant RT to the tumor bed (and regional nodes if no lymph node dissection was performed) would have been recommended for all of these patients [23]; indeed, 74% of patients received radiation in addition to surgery [22]. This suggests that, in the modern staging era, with the appropriate application of adjuvant RT to the primary site and involved nodes, patients with stage III disease likely have a superior prognosis than was previously believed (Table 1). Indeed, ADMEC-O reports that the low event rate (in addition to lower utilization of adjuvant RT in the treatment arm) likely contributed to the negative study findings, despite a promising numerical advantage to adjuvant immunotherapy.

## 4. Radiation and Adjuvant Immunotherapy

In the metastatic setting, MCC has an excellent response rate to checkpoint immunotherapy. One recent study found that, in a subset of immunotherapy-naïve patients, dual-agent ipilimumab–nivolumab achieved a 100% response rate, with only 9% progressing at 14.6 months median follow-up; in that study, the addition of radiation with the intention of promoting an abscopal reaction did not show an effect, though this may warrant further exploration in other treatment settings [10]. However, even in patients with a complete response to immunotherapy, relapses after discontinuing therapy remain an issue [44]. As a consequence, it is unclear what the optimal duration of adjuvant immunotherapy would be in the resectable, node-positive, non-metastatic setting. The initial ADMEC study randomized patients to observation or 3 months of adjuvant ipilimumab, and was closed early due to futility, with an 80% adverse event rate in the immunotherapy arm. This was published as an abstract only and grading of toxicity was not specified [45]. The subsequent ADMEC-O [22] (nivolumab) and the ongoing STAMP [46] (pembrolizumab) and ADAM [47] (avelumab) trials all used a 1-year course of adjuvant therapy.

The only completed, randomized study published to date is ADMEC-O, a multicenter randomized Phase II study of patients with fully resected MCC (62% node-positive) assigned to adjuvant immunotherapy or observation (Table 2). Both arms had a high rate of completion or elective lymph node dissection (approximately 80% among node-positive patients). The study was originally powered based on a primary endpoint of disease-free survival (DFS), with an expected 2-year DFS of 60%. However, the authors observed a 2-year DFS in the observation arm of 73% and 67% in the node-positive subgroup. At study completion, the 2-year DFS was 84% in the adjuvant therapy arm and 80% in the node-positive subgroup, compared with 73% and 67%, respectively, in the control arm. These differences were not statistically significant. In both arms, the pattern of failure was largely locoregional (11.7%), with only 5.6% of patients experiencing distant failure.

In the interpretation of these results, the authors report a higher rate of adjuvant RT utilization in the control arm (74%) versus the experimental arm (50%). Without having published data on the stage-specific use of adjuvant RT and without a pre-specified analysis of the effect of adjuvant RT, it is difficult to be certain, but these findings do imply that (1) local consolidation is critical for adequately staged node-positive MCC, (2) immunotherapy may offer a locoregional control benefit, and (3) imbalances in adjuvant RT likely reduced the observed treatment effect from adjuvant immunotherapy. Each of these considerations will be crucial in interpreting the results of the forthcoming STAMP and ADAM trials.

STAMP seeks to randomize 280 patients with stage I–III MCC to adjuvant pembrolizumab or observation without placebo blinding, but allowing patients to initiate adjuvant RT after treatment assignment [46]. Given that RT is used less often in the US for node-positive MCC [48] than in the German ADMEC studies [22], there could be a large gap between the immunotherapy and control arms of this study with respect to the utilization of adjuvant RT. While there are planned secondary analyses of RT outcomes, the primary endpoint could still be impacted by differential RT utilization. Additionally, it is possible that delays in the time to initiation of RT due to time spent in trial enrollment/randomization could attenuate the efficacy of RT [49].

In contrast, ADAM randomized 100 patients to avelumab or placebo after completion of local therapies (i.e., surgery with or without adjuvant or definitive RT) in patients with known node-positive disease. Given the omission of early stage patients with a relatively low risk of recurrence [50], the sequencing of randomization, and placebo blinding, this approach will likely yield a cleaner answer to the immunotherapy question, even if it is unlikely that this will clarify the role of adjuvant RT in node-positive MCC. However, if ADAM does not achieve similar results to the ADMEC-O node-positive immunotherapy arm, it is likely that RT utilization will be one of the first points of scrutiny.

Finally, if, in contrast to the ADMEC and ADMEC-O trials [22,45], STAMP and ADAM demonstrate improvements in DFS with immunotherapy in node-positive MCC, it is likely that there will still be a role for adjuvant RT. First, given the 2-year recurrence rate of 20%, which is driven predominantly by local–regional failure in ADMEC-O, either avelumab or pembrolizumab would have to demonstrate an excellent drug-specific benefit over the nivolumab results in a trial where half of those patients received adjuvant RT. Second, the toxicity and cost of a year of adjuvant immunotherapy therapy may outweigh its benefits (ADMEC reported 80% toxicity and ADMEC-O reported 42% G3-4 toxicity) [22,45]. Finally, immunosuppressed patients comprise about 10% of patients with MCC and typically have a more aggressive disease course, but are largely excluded from both ADAM and STAMP [51,52,53].

## 5. Radiation and Neoadjuvant Immunotherapy

Based on similar data and goals for controlling microscopic distant disease, the CheckMate 358 study recruited MCC patients with stage IIA–IIIB disease with a primary tumor of ≥ 2 cm and treated them with two cycles of neoadjuvant nivolumab prior to surgery (Table 2). Twenty-six of thirty-nine (66%) patients had node-positive disease, similar to ADMEC-O (62% node-positive) [20,22]. At 2 years of follow-up, they observed a recurrence-free survival (RFS) of 68.5%, but with a significant difference based on the pathologically complete response (pCR). Patients with a pCR had a 2-year RFS of 88.9% (no patients had a recurrence before 18 months), whereas those without a pCR had a 2-year RFS of 52.2%. Only 7.7% of patients experienced grade 3–4 toxicity related to treatment.

Putting these results in context is difficult; however, because 17 patients (47.2%) had a pathologically complete response, it is possible that this cohort had a higher rate of microscopic, subclinical nodal disease that would have been detectable only by SLNB. While the overall result is excellent in comparison with some historical data, it does not appear to significantly outperform the observation arm of ADMEC-O (2-year RFS of 73.0%) [22]. However, in CheckMate 358, only 8 patients (21%) received adjuvant RT, and in particular, only 3 of the 19 non-pCR patients. This raises the question of whether higher utilization of adjuvant RT in the non-pCR subgroup would have improved the outcomes in these patients who experienced a high rate of recurrence. Additionally, the fact that the overall cohort did not outperform the ADMEC-O control arm, despite the excellent RFS in the pCR group (which represented about half of the cohort), also adds to concerns that delays to surgery could be deleterious, particularly for patients who do not have a favorable response to neoadjuvant immunotherapy. The authors reported that three patients (7.7%) did not undergo surgery due to progression on neoadjuvant therapy or from treatment-related toxicity.

There are three small ongoing studies that will further explore neoadjuvant therapy. The phase 2 MERCURY trial in Italy is recruiting 36 patients to receive one cycle of neoadjuvant retifanlimab combined with etoposide and cisplatin (or carboplatin in patients unfit for cisplatin), which will be followed by adjuvant RT as indicated [54]. In the US, the University of Pennsylvania is conducting a phase 2 study in 15 patients with stage I–III MCC with a single neoadjuvant dose of pembrolizumab followed by a year of adjuvant pembrolizumab postoperatively [55]. Similarly, at Moffitt, 26 patients with resectable stage II–IV MCC will receive two cycles of lenvatinib and pembrolizumab neoadjuvantly followed by resection with or without adjuvant RT, followed by one year of pembrolizumab [56] While these studies will help to improve the estimation of the pCR rate and the likelihood of delays or complications rendering patients inoperable with neoadjuvant immunotherapy, they do not have control arms. The inclusion of adjuvant therapy in several of these studies also will make it difficult to assess the prognostic value of achieving a pCR and could result in overtreatment, based on the early data from CheckMate 358.

## 6. Challenges to Concurrent Immunotherapy and Radiation

In considering the future of adjuvant radiation in the context of immunotherapy potentially entering the locally advanced MCC setting, the theoretical immunomodulatory effects of radiation may need further exploration. Recent negative studies of immune checkpoint inhibitors in combination with (chemo)radiation including large elective nodal volumes [57,58,59] have led some to hypothesize that the irradiation of nodal basins may interfere with the immune response. There are ongoing smaller studies on combination radiation and immunotherapy, which are exploring small-volume radiation with stereotactic ablative radiotherapy [60]. However, it is also possible that there are drug-specific limitations and that optimal combination regimens will be identified with further exploration [61]. A recent counter-example to the hypothesis of nodal irradiation interfering with immunotherapy includes the KEYNOTE-A18 trial in locally advanced cervical cancer, which found that the addition of pembrolizumab to chemoradiation (which included pelvic nodal irradiation) improved progression-free survival [62]. In short, the potential negative effects of radiation on immunotherapy remains to be clinically proven.

The STAMP trial may offer some insight into the clinical relevance of the interaction between radiation and immunotherapy in MCC. Patients are allowed to receive radiation either prior to starting pembrolizumab or concurrently with pembrolizumab. It remains to be seen if there will be enough patients in these groups for a comparison. Given that concurrent radiation with checkpoint immunotherapy is generally safe [63], efficacy and individual patient factors will likely be crucial to decision-making for how to incorporate these two modalities, if adjuvant immunotherapy is found to be effective in the node-positive, non-metastatic MCC indication.

## 7. Conclusions

Overall, the landscape of node-positive MCC is rapidly evolving, with significant efforts being made to study adjuvant immunotherapy, and some smaller-scale investigations of neoadjuvant immunotherapy. It remains to be seen to what degree the excellent results in the ADMEC-O study were the result of stage migration, high utilization of adjuvant RT, or random factors in a moderately sized cohort of patients.

In the era of modern staging, the most common site of disease recurrence in resected, node-positive MCC is locoregional. This favors the continued use of adjuvant RT, which permits smaller surgical margins (especially in the setting of primary tumors of the head and neck), and treatment of nodal basins not amenable to SLNB or dissection. However, given the often complex anatomical differences presented by each patient’s case, as well as the frequent multi-morbidity and variable performance status of patients, it is likely that individualized multidisciplinary decision-making will remain an important component of MCC care. Ongoing clinical trials will yield insights into which patients are at low risk of recurrence in the immunotherapy era, but formal studies are necessary to determine if adjuvant RT could be routinely omitted in these patients.

## 8. Future Directions

Ongoing trials will elucidate the efficacy and toxicity of adjuvant immunotherapy, but several key methodological limitations pose persistent questions as to whether adjuvant RT could be eliminated in select node-positive patients. The low utilization of adjuvant RT (e.g., the ADAM trial, which randomizes patients after the completion of local therapy), could exaggerate the effectiveness of immunotherapy, leaving the benefit in patients who systematically receive RT uncertain. Conversely, imbalances in the utilization of RT (e.g., the STAMP trial, where the decision for adjuvant RT can be made after unblinded treatment assignment) could attenuate or otherwise confound the effects of immunotherapy. If adjuvant immunotherapy offers a benefit that is not additive to radiation (which is possible given the predominantly locoregional pattern of failure in contemporary data), it will be important to consider the relative costs and toxicity profiles of these two modalities as guidelines and patient-level treatment decisions evolve.

Neoadjuvant therapy offers a particularly interesting opportunity for future study, as patients with pCR appear to have high long-term disease-free survival, while patients without pCR appear to be at risk of rapid disease recurrence. A randomized study of surgery with adjuvant radiation versus short-course neoadjuvant immunotherapy followed by surgery with risk-adapted adjuvant RT (i.e., observation in pCR patients versus adjuvant RT in patients without pCR) may result in a treatment paradigm with a shorter, less toxic, and less expensive [64] immunotherapy regimen, while also eliminating the need for adjuvant RT in a meaningful proportion of patients.

To date, clinical trials of immunotherapy in node-positive, non-metastatic MCC have not found compelling evidence to eliminate adjuvant RT. Secondary analyses of ongoing immunotherapy trials may improve our understanding of the magnitude of the benefit of adjuvant RT and help to guide future developments in this field.

## Figures and Tables

**Table 1 cancers-15-05550-t001:** Case series of adjuvant radiation including node-positive patients.

Paper	N	Node-Positive (%)	SLNB Performed (%)	Radiation			Median Follow-Up	Recurrence Endpoints					
				Adjuvant	Definitive	Total N (%)		Reported Endpoint by Stage	I	II	III a	III b	Total
Santamaria-Barria et al. [6]	161	32%	17.0%	79	0	79 (49%)	36 months	Recurrence rate (%)	34	50	47	68	44
Harrington et al. [7]	179	23%	3.4%	95	37	132 (74%)	48 months	Locoregional recurrence (%)	-	-	-	-	8.3
Hui et al. [8]	176	35%	4.0%	-	-	165 (94%)	26 months	Locoregional recurrence (%)	-	-	-	-	35.0
Fields et al. [11]	153	29%	100%	30	-	30 (20%)	41 months	Locoregional recurrence (%)	-	-	-	-	13.0
Veness et al. [5]	86	41%	18.3%	43	6	49 (57%)	31 months	Local and/or nodal relapse (%)	-	-	-	-	31.0
Fang et al. [12]	50	100%	52.00% (remainder cN+)	45	2	47 (94%)	18 months	2-year regional recurrence-free survival (%)	-	-	100	75	12.0
Mehrany et al. [13]	60	33%	100%	16	-	16 (27%)	SLNB (−) 7.3 months SLNB (+) 12 months	Local, regional, or distant recurrence (%)	-	-	-	-	33.0
Deneve et al. [14]	38	100%	0.0%	26	9	35 (92%)	25.1 months	Regional recurrence (%)/any recurrence (%)	-	-	-	11/34	-
Ma et al. [15]	962	100%	100%	356	-	356 (37%)	32.0 months	5-year overall survival (%)	-	-	50.9	-	50.9
Cass et al. [16]	80	32.5%	80%	71	8	79 (99%)	35 months	3-year local recurrence-free survival (%)	-	-	-	-	94
Broida et al. [17]	120	31%	91%	86	-	86 (72%)	48 months	5-year disease-specific survival (%)	88		89	40	82
Pottier et al. [18]	312	28.9%	37%	226	-	226 (72%)	-	Recurrence rate (%)	37.6		22.2	50.7	39.5
Strom et al. [19]	113	26.1%	61.1%	81	-	81 (72%)	27 months	3-year recurrence-free survival (%)	83.5		87.4	50.0	-

Case series of MCC reporting recurrence rates and including node-positive patients with use of adjuvant radiation therapy. Abbreviations: N = number, SLNB = sentinel lymph node biopsy, cN+ = clinically node-positive.

**Table 2 cancers-15-05550-t002:** Radiation in clinical trials of immunotherapy in node-positive patients.

**Neoadjuvant Immunotherapy**										
**Paper**	**N**	**Node-Positive (%)**	**SLNB Performed (%)**	**Radiation**				**Recurrence Endpoints (%)**		
				**Adjuvant**	**Definitive**	**Total N (%)**	**Median Follow-Up**	**Reported Endpoint**		**Total**
Topalian et al. [20]	39	66	-	8	1	9 (23%)	20 months	2-year recurrence rate (%)		32.5
**Adjuvant Immunotherapy**										
Paper	N	**Node-Positive (%)**	**SLNB Performed (%)**	**Radiation**				**Recurrence Endpoints (%)**		
				**Adjuvant**	**Definitive**	**Total N (%)**	**Median Follow-Up**	**Reported Endpoint**	**Study Arm**	**Total**
Becker et al. [22]	179	62	68 *	104	0	104 (58%)	24.3 months	2-year disease-free survival (%)	Immunotherapy arm	84
									Observation arm	73
Becker et al. [45]	40	**	**	**	**	**	22.3 months	Hazard ratio of recurrence	1.80 (95% confidence interval 0.3–10)	

Trials of immunotherapy in non-metastatic MCC reporting recurrence rates and including node-positive patients. Abbreviations: N = number, SLNB = sentinel lymph node biopsy. * study protocol suggests all patients should receive pathologic nodal staging; authors report 49% of patients receiving “completion or elective” lymph node dissection. ** The ADMEC study was terminated due to futility analysis, results are only available in limited abstract format.

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
