# Peer review of "Adjuvant Radiation in Resectable Node-Positive Merkel Cell Carcinoma in the Immunotherapy Era: Implications for Future and Ongoing Trials"

_cancers, 2023, doi:10.3390/cancers15235550_

Round 1

Reviewer 1 Report

Comments and Suggestions for Authors

Overall this review provides a valuable perspective in a rapidly evolving field. I think publication of this review will be helpful for the multidisciplinary merkel cell carcinoma care team in thinking about how different modalities of therapy can be combined as systemic therapy and surgery evolves. A few areas that need work:

Table 1 focuses on older papers that the largely predate the immunotherapy era. Given the focus of this review I would suggest adding the following papers to this Table (and corresponding references):

PMID 37106277, 37659660, 37504322, 37307995, 27300153. 

Line 64 notes immunotherapy toxicities being "chiefly cutaneous/fatigue" but I don't think these are the toxicities most patients or Med Oncs worry about which makes "chiefly" perhaps not the best wording.

Line 106 notes the additive risk of lymphedema of RT beyond SLNB is 5% but this is really only relevant for the axilla (as noted from breast literature). If extrapolating from other diseases, probably worth noting inguinal nodal RT risk after SLNB (from GROINS-V-II) which is notably higher.

In the Conclusion/Future Directions would acknowledge that RT is not always less morbid than surgery (especially depending on target/dose/fractionation). Would think this nuance needs more discussion, especially uncertainty about optimal dose/fractionation as well as the decision of surgery vs. RT vs. both as local therapy likely being best made in multiD setting taking into account particular morbidity for that specific patient.

Reviewer 2 Report

Comments and Suggestions for Authors

In this review article, Riviere and colleagues address the role of adjuvant radiotherapy (RT) in the context of multimodal concepts with immune checkpoint inhibitors (ICI) for the treatment of resectable node-positive merkel cell carcinoma (MCC). The authors examine current studies on this topic to clarify whether adjuvant RT after immunotherapy reduces the risk of recurrence in MCC. The clarification of this question could be decisive for the treatment of future MCC patients and, for example, reduce or eliminate the risk of late effects of RT if it is omitted. However, the data available to date do not support the omission of RT, but the authors also provide important information and recommendations for the design of future studies on this issue.

Together, the authors present a well-structured and easy-to-read manuscript that summarizes the most important current studies on this topic.

Just a small note from my side that should be considered in a minor revision:

Concerning RT-ICI combinations:

Does the immunostimulatory effect of RT, which can promote synergies with ICI, also play a role in MCC? Are the systemic (abscopal) effects on distant metastases relevant here?

On the other hand, the immunosuppressive effect of RT due to radiation damage to the hematologic system is also currently being discussed. This leads to a lower efficacy of ICI due to the irradiation of tumor-draining lymph nodes and radiation-associated lymphopenia.

I would appreciate a short paragraph on this topic and its relevance to the tumor entity of MCC.

Reviewer 3 Report

Comments and Suggestions for Authors

The Authors performed a study about Adjuvant Radiation in Resectable Node-Positive Merkel Cell  Carcinoma in the Immunotherapy Era. The article is generally of interest and well writte. I suggest only some minor changes:

- Please change and divide Table 1 according to adjuvant and neo-adjuvant therapy, in order to divide studies about adjuvant and studies about neo-adjuvant therapies.

- If possibile, please add some cases from your Institute/Department, reporting and higlighitng your experience about this interesting topic.

Thank you.

Reviewer 4 Report

Comments and Suggestions for Authors

This manuscript deals with the current (neo)adjuvant treatment in Merkel cell carcinoma (MCC), comparing the effects of adjuvant radiation therapy (RT) with those of adjuvant immunotherapy. The authors are clearly experts in the field of MCC and well familiar with current clinical trial in this field. Indeed, it is a relevant and still unanswered question whether adjuvant RT will remain standard of care when adjuvant immunotherapy might become a new therapeutic standard. However, the manuscript is less of a review (as classified) but more a discussion of the different clinical trial concepts and potential conclusions that may be drawn from them. This makes it quite difficult for readers to follow the authors´ considerations, which also contain a large amount of speculation. Thus, the reviewer suggests that the authors spearate the review of the various clinical trials they refer to (adjuvant immunotherapy and adjuvant RT trials - trial design and outcome, no own conclusions or speculations) from a discussion of these data.

Moreover, Table 1 needs further editing, as several aspects remain unclear or undefined (e.g. what is meant by recurrence endpoints I-IIIb? Is the recurrence rate referring to the node positive or to all patients? Why is it relevant whether a PET/CT was done ? Is the outcome different with respect to “adjuvant” ws. “definitive” radiation ? What are the respective radiation doses ?). As the reported trials differ significantly in their designs, the reviewer is uncertain whether it is suitable to summarize them in a joint table. If a table of the RT trials will remain in the manuscript, another table that refers to the adjuvant immunotherapy trials should also be included.

Round 2

Reviewer 4 Report

Comments and Suggestions for Authors

Thank you for re-editing the manuscript, I have no additional comments or requests